# Effect of Additives Inclusion in Gilthead Seabream (*Sparus aurata* L.) Diets on Growth, Enzyme Activity, Digestibility and Gut Histology Fed with Vegetable Meals

**DOI:** 10.3390/ani13020205

**Published:** 2023-01-05

**Authors:** Glenda Vélez-Calabria, Ana Tomás-Vidal, David S. Peñaranda, Miguel Jover-Cerdá, Silvia Martínez Llorens

**Affiliations:** Research Group of Aquaculture and Biodiversity, Institute of Animal Science and Technology, Universitat Politècnica de València, Camino de Vera 14, 46022 València, Spain

**Keywords:** gilthead seabream, additives, hydrolyzed porcine mucosa, nucleotide concentrates

## Abstract

**Simple Summary:**

Aquaculture is currently directing its efforts towards the principles of sustainability, especially in the reduction in the use of fishmeal, which is why it faces a constant search for alternative sources. In the current work, it has been found that the use of two different feed additives, hydrolyzed porcine mucosa and nucleotide concentrate, in different percentages in a plant-based diet for gilthead seabream, improves growth and feed efficiency.

**Abstract:**

The fishmeal replacement by vegetable meals or other alternative sources, without affecting fish performance and productivity, is one of the principal challenges in aquaculture. The use of hydrolyzed porcine mucosa (HPM) and nucleotide (NT) concentrates, as feed additives in gilthead seabream (*Sparus aurata* L.) non-fishmeal diets was assessed in order to determine the possible effects on growth, feed efficiency, protein digestion, and gut histology when these were included in a plant-based diet (HPM 1% and 2%, P1 and P2; NT 250 and 500 ppm, N250 and N500), in comparison with two control diets, AA0 (100% plant-protein-based diet) and FM100 (100% fishmeal-protein-based diet). Diets were assayed in triplicate and the growth assay lasted 134 days. Results showed a significant improvement in all groups in terms of final weight and specific growth rate in comparison with the AA0 group. An improvement in the feed conversion ratio and the protein efficiency ratio was also observed when the additives were included in lower percentages (P1 and N250) compared to the FM100 group. Significant differences were found in hepatosomatic index, villi thickness, and goblet cells. Thus, the inclusion of NT and HPM was tested as beneficial for the improvement of efficiency of plant feed in seabream.

## 1. Introduction

The fishmeal replacement (FM) by alternative sources in diets for gilthead seabream (*Sparus aurata* L.) is necessary to ensure aquaculture sustainability. Many researchers have focused on finding new ingredients, such as a wide variety of vegetable meals, that can substitute FM in fish feed.

Good growth results have been obtained for gilthead seabream when 50% [1] and 75% [2] of the FM was replaced with mixtures of vegetable meals, although histological alterations in the gut were observed when plant ingredients were included in percentages above 60% [3], and also an immunosuppressive effect when the level of inclusion exceeded 75% [4]. Kissil and Lupatsch [5] replaced 100% of the FM by a mixture of high-quality vegetable sources and the addition of synthetic amino acids, but commercial application of the designed feed was unfeasible. Feed formulation with vegetable ingredients was limited by the protein content of different sources, the presence of antinutritional factors [6], and the balance of amino acids (AA).

Feed additives are added during feed preparation to improve not only the quality of the feed but also to improve the growth performance and health performance of the fish. Most feed additives are non-nutritive and include antioxidants, immunostimulants, probiotics, prebiotics, enzymes, microalgae, among others. Prebiotics are non-digestible feed ingredients that beneficially affect the host by selectively stimulating the growth or activity of one or a limited number of bacterial species already resident in the gut, and thus attempt to improve the host’s health [7]. Prebiotics have been studied in fish since 1995 [8] and different effects, such as the increase in the growth rate, the stimulation of the immune system, changes in the gut microbiota, and modulation of the gut morphology have been described [9,10,11,12,13].

Nucleotides (NTs) are low-molecular-weight intracellular compounds which play key roles in nearly all biochemical processes [14] and are the building blocks of nucleic acid (DNA and RNA). A nucleotide consists of a nitrogenous base, a sugar (ribose or deoxyribose), and one to three phosphate groups. When the phosphate group of the NT is removed by hydrolysis, the structure remaining is nucleoside. Dietary supplementation of NTs or nucleosides has been shown to benefit many mammalian physiological and nutritional functions [15,16,17,18] and both are considered as functional nutrients [19,20,21]. Although NTs can be synthesized endogenously from purines and pyrimidines originating from salvage (recycling from dead cells) and de novo synthesis (from AA) pathways in animals, it has been increasingly demonstrated that NT supplementation is required in diets for maximum production performance and immunocompetence of aquatic animals [22,23]. Moreover, (non-fishmeal) aquafeed ingredients contain relatively low amounts of nucleotides [24]. NTs participate in cell construction, tissue growth, development, and repair, fast growth, health benefits, and stress tolerance in fish, which have been regarded as “semi-essential nutritional components” or “conditional nutrition” [20,21]. NT concentrates (prebiotics) have been frequently tested in mammals, and beneficial effects on immune system [25], gut morphology [26], digestive microbiota [17], fat metabolism [27], and resistance to diseases [28] were observed many years ago. The roles of NTs and metabolites in fish diets have been sparingly studied for over 25 years [29]. NTs currently used in aquaculture [30,31] are the same as those added in infant formulae [32], including five types of nucleotide 5′-monophosphate at sodium salts, water soluble cytidine monophosphate, uridine monophosphate, AMP, inosine monophosphate (IMP), and GMP [33]. The addition of NT concentrates in feed has been tested in different species, such as rainbow trout (*Oncorhynchus mykiss*, [34]), Atlantic salmon (*Salmo salar*, [31]), red drum (*Sciaenops ocellatus*, [35]), Pacific white shrimp (*Litopenaeus vannamei*, [36]), meagre (*Argyrosomus regius*, [37]), red seabream (*Pagrus major*, [20]), and grass carp (*Ctenopharyngodon idellus*, [38]), with a variable effect on growth and survival rate. The information on the effect of NTs on feed efficiency, metabolism, and mitochondrial enzyme complexes for gilthead seabream is scarce. Couso et al. [39] described an increase in the survival rate of seabream juveniles fed with a feed which incorporated an NT concentrate, after they were exposed to the pathogenic bacteria *Photobacterium damselae* subsp. *piscicida*, which suggests an immunostimulant effect of the NT concentrate. El-Nokrashy et al. [40] found that dietary NTs supplemented at 250 mg/kg or 500 mg/kg enhanced the final body weight, weight gain, and specific growth rate of gilthead seabream, either with a dietary level of 25% FM diet or a non-fishmeal diet.

On the other hand, hydrolyzed porcine mucosa (HPM) is derived from the mucosa of the cleaned small intestine of pigs, and is a byproduct of the production of the anticoagulant drug heparin sodium after high-temperature spray drying [41]. To increase the yield of heparin sodium, enzymes are added during the extraction process for the reaction [42]. Therefore, HPM contains polypeptides, oligopeptides, small peptides, and free AA [43,44]. As a new type of functional nutritive animal protein source, HPM has been widely used in lactating sows [45] and post weaning pigs [43,46,47] owing to its high protein content, balanced AA composition, and high safety profile [41,44]. There is little research on the use of HPM in aquatic animals. To the best of our knowledge, only one study on carp (*Cyprinus carpio*) has been reported thus far [48], which showed that replacing FM with 3% HPM equivalents had no significant effect on the growth performance of carp; however, there was a significant reduction in intestinal fold depth and villi height.

The current work assesses the effects of including, in vegetable feeds for fish, an NT concentrate and HPM in different doses. Growth, nutrient efficiency, biometric indices, dry matter, protein and AA digestibility, and intestinal histology have been evaluated and compared to values obtained in fish fed with FM.

## 2. Materials and Methods

The experimental protocol was reviewed and approved by the Ethics and Animal Welfare Committee of the Universitat Politècnica de València (Official bulletin No. 80 of 06/2014), following Royal Decree 53/2013 and the European Directive 2010/63/EU on the protection of animals used for scientific research, with the purpose of minimizing the suffering of animals.

### 2.1. Rearing System

The trial lasted 134 days and was conducted in 18 cylindrical fiberglass tanks (1750 L) within a recirculating saltwater system (75 m^3^ capacity) with a rotary mechanical filter and a gravity biofilter (6 m^3^ capacity). All tanks were equipped with aeration, and the water was heated with a heat pump installed in the system. The water temperature was 22.0 ± 0.52 °C, salinity was 30 ± 1.7 g L^−1^, dissolved oxygen was 6.5 ± 0.49 mg L^−1^, and pH ranged from 7.5 to 8.5. The photoperiod was natural, and all tanks had similar lighting conditions.

### 2.2. Fish

Gilthead seabream were obtained from the fish farm PISCIMAR in Burriana (Valencia, Spain) and after two weeks of acclimation to laboratory conditions, fed a standard commercial diet (480 g kg^−1^ crude protein, CP; 230 g kg^−1^ crude lipid, CL; 110 g kg^−1^ ash; 22 g kg^−1^ crude fiber, CF and 140 g kg^−1^ nitrogen-free extract, NFE), and were distributed in 18 cylindrical fiberglass tanks (three per treatment) in groups of 25 fish in each tank. The average weight of fish was 11 ± 1.2 g at the initiation of the experiment.

### 2.3. Diets and Feeding

Diets were prepared as pellets by extrusion cooking with a semi-industrial twin-screw extruder (CLEXTRAL BC-45, Firminy, St Etienne, France) located at the Universitat Politècnica de València. The processing conditions were as follows: a temperature of 110 °C, a pressure of 40–50 atm, and a screw speed of 100 rpm.

In the vegetable diet (AA0), the protein fraction was supplied entirely by commercial plant ingredients, complying with the minimum requirements of essential amino acids (EAA) established by Peres and Oliva-Teles [49]. Additives were assayed in different doses: nucleotide concentrate in doses of 250 ppm (N250) and 500 ppm (N500), and HPM at levels of inclusion of 1% (P1) and 2% (P2), replacing wheat gluten. All these diets were isonitrogenous and isoenergetic and had the same chemical composition. In the control diet (FM100), all the protein was provided by FM. An amount of 50 g kg^−1^ of chromium oxide (Cr_2_O_3_) was used as an inert marker for the digestibility trial in these same diets. All dry ingredients were uniformly mixed together before adding the liquid ingredients (vegetable and fish oils). Ingredient content is shown in Table 1.

Each experimental diet was assayed in three different tanks, randomly assigned. Fish were handfed three times per day (8:00, 12:00, and 16:00) to apparent satiation in a weekly feeding regimen of six days and one of fasting. Pellets were distributed slowly, permitting all fish to eat. Fish were observed daily in tanks and were weighed individually every four weeks, using clove oil containing 87% eugenol (Guinama^®^, Valencia, Spain) as an anesthetic (1 mg per 100 mL of water) to minimize their suffering, in order to evaluate fish growth along the assay, determine growth parameters, and assess their health status.

### 2.4. Proximate Composition and Amino Acids Analysis

Chemical analyses of the dietary ingredients were determined prior to diet formulation. Diets and their ingredients were analyzed according to AOAC [50] procedures: dry matter (105 °C to constant weight), ash (incinerated at 550 °C to constant weight), crude protein (N × 6.25) by the Kjeldahl method after an acid digestion (Kjeltec 2300 Auto Analyser, Tecator, Höganas, Sweden), crude lipid extracted with methyl-ether (Soxtec 1043 Extraction Unit, Tecator), and crude fiber by acid and basic digestion (Fibertec System M., 1020 Hot Extractor, Tecator). All analyses were performed in triplicate.

Following the method previously described by Bosch et al. [51], AA of diets were analyzed through a Waters HPLC system (Waters 474, Waters, Milford, MA, USA) consisting of two pumps (Model 515, Waters), an auto sampler (Model 717, Waters), a fluorescence detector (Model 474, Waters), and a temperature control module. Aminobutyric acid was added as an internal standard pattern before hydrolyzation. The AA were derivatized with AQC (6-aminoquinolyl-N-hydroxysuccinimidyl carbamate). Methionine and cysteine were determined separately as methionine sulphone and cysteic acid after oxidation with performic acid. AA were separated with a C-18 reverse-phase column Waters Acc. Tag (150 mm × 3.9 mm), and then transformed to methionine and cystine. Proximate composition and EAA content of experimental diets is shown in Table 2.

### 2.5. Growth and Nutrient Efficiency Indices

The growth and nutrient efficiency indices were determined at the end of the experiment and the tank was used as an experimental unit. The specific growth rate (SGR), feed intake (FI), feed conversion ratio (FCR), protein efficiency ratio (PER), and survival (S) were obtained, taking into account the monthly reported biomass of dead fish.

### 2.6. Biometric Indices

At the end of the feeding trial, all the fish were individually weighed. Three fish from each tank, nine per treatment, were randomly slaughtered using a lethal bath of clove oil (150 mg L^−1^), for the determination of biometric indices. The samples from each tank were pooled and stored at −30 °C. Fish total weight and length, as well as viscera, visceral fat, and liver weights were recorded for determination of condition factor (CF), viscerosomatic (VSI), visceral fat (VFI), and hepatosomatic (HSI) indexes.

### 2.7. Digestibility

Apparent digestibility experiment was carried out in the same tanks at the end of the growth experiment. The same six diets were used but chromium oxide (50 g kg^−1^) was added as an inert marker. The fish that were not used for the analyses were left in the tanks and continued to feed with these diets. Feces collection was carried out three times a week (to minimize fish suffering) for a period of 21 days. Fish were fed once a day in the morning (05:00) and fecal collection took place 8 h later (13:00). Before feeding, fish were fasted for two days. Extraction was performed by stripping (applying pressure on the ventral region from the pelvic fins to the anus). Wet fecal content was collected and dried at 60 °C for 48 h prior to analysis.

Fecal composition of DM, CP, and AA was analyzed by the same procedure as in diets, after drying at 100 °C until constant weight. Chromium oxide was determined in the diets and feces using an atomic absorption spectrometer (Perkin Elmer 3300, Perkin Elmer, Boston, MA, USA) after acid digestion. Analysis was performed in duplicate. DM, CP, and AA of diet and feces were used to determine apparent digestibility coefficient (ADC) of DM, CP, and AA with the following formula:ADC (%) = 100 × [1 − (% marker in diet/% marker in feces) × (% N in feces/% N in diet)](1)
where N is the nutrient (DM, CP, and AA).

### 2.8. Histological Analysis

Histological analysis of the current study was performed on fish fed with the control diet AA0 (diet to improve), FM100 (diet control), and the diets with which the best results were obtained, P1 and N250. At the end of the growth experiment and before the histological analysis, the guts of three fishes per tank fed with FM100, AA0, N250, and P1 diets were divided in three portions: proximal intestine (PI), middle intestine (MI), and distal intestine (DI). PI samples were preserved in phosphate-buffered formalin (4%, pH 7.4). All of the formalin fixed tissues were routinely dehydrated in ethanol, equilibrated in ultraclean, and embedded in paraffin according to standard histological techniques. Transverse sections were cut with a thickness of 5 µm with a microtome Shandon Hypercut (five sections per paraffin block were obtained) and stained with haematoxylin and VOF (Light green, Orange G, and Fuchsin) for examination.

A total of 108 sections were analyzed under the light microscope (Eclipse E400 Nikon, Izasa S.A., Barcelona, Spain). For the measurements and observations of the intestine, we used a combination of criteria reported by several authors [4,52,53]. Daprà et al. [54] and Øverland et al. [55] use the following parameters: serous layer (SL), muscular layer (ML), submucous layer (SML), villi length (VL), villi thickness (VT), and lamina propria (LP). Six measurements per section of each parameter were performed, and average values were determined. In addition to the measurements, a quantification of goblet cells (GC) was made by counting the number of GC present in each villus. We used six villi per section.

### 2.9. Digestive Enzyme Activity

Digestive tracts of three fish per tank were sampled at the end of the assay, 134 days after initiation of the experiment. To ensure the presence of content along the whole digestive tract, fish were fed at 20:00 on the day before and at 8:00 on the sampling day.

Fish were dissected in order to obtain the digestive tract, after being anesthetized using clove oil and sacrificed by cold shock. Two different kinds of samples were considered and obtained: stomach (S) and gut (G). They were stored at −20 °C until enzymatic extraction. Enzyme extracts for protease analysis were obtained by manual disaggregation, dilution in distilled water (1 g of sample: 3 mL of distilled water), followed by homogenization by T 25 digital ULTRA-TURRAX^®^ (IKA^®^, Staufen, Germany), maintaining tubes on ice, and centrifugation at 12,000 rpm and 4 °C for 15 min. Supernatant were stored at −20 °C until enzyme analysis.

Pepsin assays were performed on S samples and total alkaline protease, trypsin, α-amylase, and alkaline phosphatase (ALP) assays were performed on G. Enzyme activities, expressed as U per g of tissue.

Acid protease (pepsin) activity was evaluated using 0.5% hemoglobin *w*/*v* as substrate in 100 mM glycine—HCl buffer, pH 2.5, at 280 nm, following the method detailed by Anson [56] and modified by Díaz-López et al. [57]. One unit of activity was defined as 1 μg of tyrosine released per min (extinction coefficient = 0.0071 mL μg^−1^ cm^−1^).

Trypsin activity was obtained by a kinetic assay using Nα-Benzoyl-DL-arginine p-nitroanilide (0.5 mM BAPNA) as a substrate in 50 mM Tris-HCl buffer containing 20 mM CaCl_2_, pH 8.2, following the method developed by Erlanger et al. [58]. The increase in absorbance at 405 nm was measured every 30 s for 5 min. One unit of activity was defined as 1 μg of p-nitroanilide released per min (extinction coefficient = 0.0637 mL μg^−1^ cm^−1^).

Total alkaline protease activity was tested using 1% casein *w/v* as substrate in 100 mM Tris-HCl buffer containing 10 mM CaCl_2_, pH 7.5, at 280 nm, following the method detailed by Kunitz [59] and modified by Walter [60]. One unit of activity was defined as 1 μg of tyrosine released per min (extinction coefficient = 0.0071 mL μg ^−1^ cm^−1^).

α-Amylase activity was determined by a kinetic assay using a commercial kit (Amylase MR, Cromatest, Linear Chemicals S.L., Barcelona, Spain), following manufacturer’s instructions. The increase in absorbance at 405 nm was measured every 30 s for 5 min, after an incubation period of 1 min. One unit of activity was defined as 1 μg of 2-chloro-p-nitrophenol released per min during the enzymatic reaction at 37 °C (Extinction coefficient = 0.0818 mL μg^−1^ cm^−1^).

The samples for alkaline phosphatase (ALP) were diluted at a ratio of 1:20. The activity of this enzyme was measured using a kinetic assay commercial kit (ALP-LQ, Spinreact, liquid, DGKC, St Esteve d’en Bas, Girona, Spain), following manufacturer’s instructions. A total of 200 µL of reagent were added to 10 µL of the diluted homogenate. The increase in absorbance at 405 nm was measured every 30 s for 5 min, after an incubation period of 1 min. One unit of activity was defined as 1 μg of p-nitrophenylphosphate released per min during the enzymatic reaction at 25 °C.

### 2.10. Statistical Analysis

The results of the different growth and nutrient indices, biometric indices, ADCs, histological measurements, and specific enzyme activities were analyzed through an analysis of variance using the statistical package Statgraphics^®^ Plus 5.1 (Statistical Graphics Corp., Rockville, MO, USA), with a Newman–Keuls test for the comparison of the means. Initial weight was used as a covariate in the analysis of growth indices. The results are shown as the mean ± standard error (SEM). The level of significance was set at *p* < 0.05.

## 3. Results

### 3.1. Growth and Nutrient Efficiency Indices

The results obtained on growth and nutrient efficiency indices are shown in Table 3. At the end of the growth period, all groups showed a significant improvement in final body weight and specific growth rate (SGR) in comparison with the control group (AA0), which showed the lowest values (87 g and 1.59% day^−1^, respectively), although slightly inferior to the FM100 diet. All diets were well accepted and no significant statistical differences between groups for feed intake (FI) and survival rate (S) were detected. The feed conversion ratio (FCR) was significantly better when the additives were added in lower percentages (P1 and N250), without significant differences compared to the FM100 diet. Similarly, the protein efficiency ratio (PER) presented values equal to and/or close to the FM100 diet when the additives were included in lower levels.

### 3.2. Biometric Indices

The values obtained in the biometric indices are shown in Table 4. No significant differences were found in CF, VSI, and VFI biometric indices, with the exception of the hepatosomatic index (HSI), which in the N500 group was the lowest value.

### 3.3. Digestibility

Apparent digestibility coefficients (ADC) of dry matter (ADC_DM_), crude protein (ADC_CP_), and amino acids (ADC_AA_) of the diets for gilthead seabream are presented in Table 5. The ADC results obtained were similar in all treatments, not appreciating differences in digestibility due to the use of additives compared to the non-fishmeal control diet (AA0).

### 3.4. Histological Analysis

The results of the measurement of each morphological parameter evaluated in the PI are shown in Table 6. No significant differences are shown, except in villi thickness (VT) and goblet cells (GC). VT was lower in the AA0 and N250 groups (101.2 and 102.9 µm, respectively). In terms of the secreting cell count (GC), there were differences between the control diet (AA0) and the N250 and P1 diets, resulting in lower GC.

### 3.5. Digestive Enzyme Activity

Activities of digestive enzymes pepsin and α-amylase were unaffected by diets composition, unlike total alkaline proteases, trypsin and alkaline phosphatase, which were affected. Fish fed the control diet AA0 had the lowest enzymatic activity values, except for α-amylase (Figure 1).

#### 3.5.1. Proteases Activity

No significant differences were found in pepsin activity in stomach samples from fish fed the different experimental diets (Figure 1a). The highest value was obtained in P1 group, followed by N250, while the activity registered in the other groups was lower. Total alkaline proteases activity showed significant differences in the gut tissue (Figure 1b). The control diets (AA0 and FM100) showed differences from each other, while the groups with additives (N250, P1, N500, and P2) did not, presenting very close values. Significant differences in trypsin activity were found, where fish fed the P2 diet showed the highest values, followed by N250, P1, N500, and AA0 diets (Figure 1c).

#### 3.5.2. Alkaline Phosphatase Activity

There were significant differences in ALP (brush border enzyme) activity. As in the activity of the total alkaline proteases, the control diets AA0 and FM100 presented differences between them. The groups with additives (P1, N250, and N500) showed similar values, with the P2 group presenting the highest value (Figure 1d).

#### 3.5.3. α-Amylase Activity

There were no significant differences observed among experimental groups when α-amylase activity was determined in gut tissue (Figure 1e). Highest average values were registered in the N500 group followed by the AA0 control group.

## 4. Discussion

The results in the present study clearly show that the inclusion of additives (HPM and NT concentrate) to a 100% vegetable diet, improves the values in all groups significantly in terms of growth performance and nutrient efficiency indices, compared to the control group (AA0). When the additives were included in low concentrations, the P1 and N250 groups showed the highest results regarding the final weight, SGR, FCR, and PER. Similarly, the N500 and P2 groups showed close values. Regarding the HPM or HPM equivalents, Yang et al. [61] found the addition of low doses of enzyme-digested hydrolyzed porcine mucosa (30 g kg^−1^) feasible in hybrid grouper (*Epinephelus fuscoguttatus* ♀ × *E. lanceolatus* ♂) feed, as there were no significant changes in growth performance in terms of SGR, PER, and FCR compared with controls. Likewise, results obtained by Gao et al. [48] suggest that dried porcine soluble, a byproduct of heparin extraction from pig intestines, can replace FM without any adverse impact on the growth performance of carp (*C. carpio*), thus confirming that additives greatly improve the use of vegetable diets. This may be due to the fact that the HPM has a high level of free AA and oligopeptides, which gives it high digestibility. With reference to NT concentrate, El-Nokrashy et al. [40] found that dietary NTs supplemented at 250 mg kg^−1^ or 500 mg kg^−1^ enhanced the final body weight, weight gain, and SGR of gilthead seabream compared to the control group. In contrast, the results obtained by Ridwanudin et al. [62] clearly showed that there was no positive effect of dietary nucleotides on fish growth of juvenile rainbow trout (*Oncorhynchus mykiss*). Commercial NT concentrates are normally a balanced concentrate of free NTs and active precursors, obtained from yeast. Thanks to these characteristics, they minimize the intestinal inflammatory response in diets with high incorporation of raw materials of plant origin, helping to enhance the digestibility of the diet and thus improve the intestinal health of animals. Generally, nucleoproteins are degraded by proteases to peptides and nucleic acids. The nucleic acids are cleaved by nucleases to NT. The phosphate groups of NTs are removed primarily by intestinal alkaline phosphatases to form nucleosides. Intestinal proteases and alkaline phosphatases are well elucidated in fishes. Sugars may be cleaved by nucleosidases to produce free purine and pyrimidine bases. The nucleosides and nitrogenous bases are absorbed by the gut mucosa. The digestion and absorption of NTs in fish are influenced by various environmental and/or physiological factors [63].

In the present study, no significant differences were found in CF, which is supported by the results obtained by Gao et al. [48] and Yang et al. [61] when they replaced FM with HPM equivalents in diets for carp and hybrid grouper, respectively. The VSI did not show significant differences either, such as in the study by Ridwanudin et al. [62], in diets with dietary NT for rainbow trout, unlike Gao et al. [48] and Yang et al. [61], who did find that the VSI increased significantly in diets that included HPM equivalents compared with the control group. The HSI presented significant differences, where diets N250, P1, and P2 showed values close to the control diet. This may be due to the direct utilization of the content of small peptides present in the feed, which increases metabolic load and liver mass [64]. The HSI of the N500 diet was significantly lower than the control diet. Similarly, Ridwanudin et al. [62] found that the effect of dietary NT significantly decreased HSI in rainbow trout. Pafundo et al. [65] found that hepatocyte size decreased in rainbow trout after NT treatment. Thus, increased NT availability in rainbow trout appeared to decrease hepatocyte size and ultimately reduced HSI.

The digestibility results obtained in the present trial were similar in all treatments, with no differences in digestibility due to the use of feed-additive NT concentrate and HPM, compared to the fishmeal-free diet (AA0). Fish fed with diets containing high concentrations of plant protein (PP) often show low feed digestibility, as a consequence of using ingredients known to be high in fiber or have very high starch content [66,67]. Supplementation of feeding stimulant, which helps to increase the digestibility of plant protein-based diet through increasing the secretion of various digestives enzymes, is also well documented [68,69,70]. The results obtained by Hossain et al. [23,71] indicate that supplementation with IMP nucleotide increased the efficiency of using soy protein concentrate (SPC ≤  75%) as the only source of protein, and that NT supplementation may be effective as a functional supplement in the diet of red seabream (*Pagrus major*). The absorption of hydrolyzed proteins can be three times faster than those that are not hydrolyzed [72]. The intestines of animals can absorb AA easier when they are in the form of small peptides, thus the incorporation of hydrolyzed proteins in diets can benefit the growth and development of animals [73], since the hydrolysis process transforms longer protein chains into smaller peptide chains and free AA. The results of the trial carried out by dos Santos Cardoso et al. [74] showed that the hydrolyzed swine mucus protein (HSMP) used at 20% in Nile tilapia (*Oreochromis niloticus*) diets has higher ADCs of protein, energy, and various AA.

In this study, it can be seen that the histological parameters SL, ML, SML, VL, and LP of PI were not significantly affected by diet, unlike VT and GC, which did present significant differences. Digestive functioning depends on the development of the intestine; better development of intestinal villus length and villus width increases the intestinal surface area for better nutrient absorption [75]. Our results indicate that in the P1 group, the VL and VT were numerically higher than that of the control group (AA0). This may be because small peptides can effectively stimulate and induce increases in brush border (BB) enzyme activity in the intestinal chorion and accelerate villus growth, promoting intestinal digestive function [76].

No significant differences were found in the LP, although the non-fishmeal diet (AA0) reached the lowest value of all. In several studies, the histological results show a circulatory disorder, with a marked infiltration of leukocytes and eosinophilic cells in the LP, when using diets with maximum replacement that imply an increase in the cellularity of the LP [53,55,77,78].

Considering the SL and ML, these did not present significant differences, resulting in the smallest thickness in both parameters, obtained in the group of the control diet (AA0) with respect to the other diets. In contrast, the results obtained by Yang et al. [61] in hybrid grouper feed indicate that the muscle thickness of PI in control group was significantly higher than in the groups with HPM equivalents addition. Normally, SL is the layer located on the outside of the ML, it is formed by a secretory epithelial layer (mesothelium) and another layer of connective tissue that supplies nutrients to the epithelium through blood vessels [79]. The ML is the thinnest part of the digestive tract in the distal portion. The ML is the area where much of the nutrient uptake occurs, but virtually no mechanical processing occurs [80].

The SML no showed significant differences in the different experimental diets, the N250 diet being the lowest. Dietary NTs are a group of additives that are widely used in aquaculture as feed attractants. They are often implicated in numerous positive physiological effects including increased growth performance, feed utilization, and enhanced intestinal fold morphology in several fish and shellfish species [23,31,81]. In addition, NTs were shown to have a protective effect in overcoming intestinal and inflammatory reactions induced by plant-rich protein diets [37]. SML is a layer composed of connective tissue, that controls the expansion of the intestine when substantial food is consumed in carnivorous fish [82], and is also responsible for most of the absorption. It has numerous blood vessels, which make it possible to allow oxygen and nutrients to reach all the cells and also remove waste material [83].

Regarding the number of CG, significant differences were observed, and their number was lower in the N250 diet compared to the control diet (AA0). A similar behavior of CG decrease was found by Valente et al. [84] when they included NTs as an additive in diets for European seabass (*Dicentrarchus labrax*). The GC, present along the entire intestine, are responsible for the synthesis and secretion of the protective mucus layer that covers the epithelium surface. This mucus layer acts as a medium for protection, lubrication, and transport between the luminal contents and the epithelial lining, and it is an integral structural component of the intestine [85,86].

The capability of fish to digest and use nutrients depends on some factors: digestive enzymes [87], the size of the fish, as well as the length of the intestine. The diet composition and the raw ingredients used in it can modulate the intestinal enzymatic profile [53], while the activity of these enzymes in the digestive tract can be utilized as an indicator of digestive capacity (the ability of fish intestine to take advantage of different nutrients) and the nutritional status of the fish [88]. In this study, the digestive enzymatic activities of the proteases presented the best results in fish fed the diets whose percentage of inclusion of additives (NT concentrate and HPM) was lower. It is worth emphasizing again that there were no major differences between the diets with additives (P1, P2, N250, and N500), while the control diets (AA0 and FM100) did present differences between them. Pepsin activity did not show significant differences, although the values of diets P1 and N250 were numerically higher than that of the control diet (AA0). The results found in trypsin activity coincide with those obtained by Yang et al. [61], in diets with low doses of HPM equivalents for hybrid grouper. It is well documented that feeding stimulants increase the secretion of different digestive enzymes. Morimoto Kofuji et al. [68] reported that supplementation of feeding stimulants (Alanine, proline, and IMP mixtures) increases the pepsin, trypsin, and chymotrypsin secretions in yellowtail (*Seriola quinqueradiata*). In white shrimp (*Litopenaeus vannamei*, [89]) and seabass (*D. labrax*, [90]), ingesting feed with an appropriate number of small peptides can increase digestive enzyme activity in the intestine and liver. Active peptides can directly act as neurotransmitters to indirectly stimulate intestinal hormone receptors or promote enzyme secretion, and can also provide a complete nitrogen framework for rapid synthesis of digestive enzymes in the body [91].

Regarding the activity of the α-amylase enzyme, no significant differences were found, but the results indicate that the control diet (AA0), fishmeal-free diet, was numerically higher than the diets with inclusion of HPM (P1 and P2). Similarly, Yang et al. [92] found that amylase activity was significantly higher in the control group (HPM0) than in groups with 3, 6, and 9% HPM inclusion in the low-fishmeal feed for hybrid groupers (*Epinephelus fuscoguttatus* ♀ × *E. lanceolatus* ♂). Amylase activities of tissues and intestinal contents vary among species and appear higher in herbivorous and omnivorous fish than in carnivorous fish [93,94,95,96]. In a number of fish species, activities of intestinal α-amylase correlate positively with dietary carbohydrate level and feeding intensity [97]. It is known that the plant carbohydrates may be well digested if the cellular membrane is partially broken, exposing the content of cells to the digestive enzymes. In seabream, this initial process of hydrolysis may be ensured by the acid environment of the stomach [98]. Further hydrolysis of starch is completed in the intestine by the action of amylase [99]. In contrast to mammals, where amylase is produced by salivary and pancreatic cells, the only source of α-amylase in fish appears to be the exocrine pancreas [100]. Previous studies have shown that carnivorous marine fish species (redfish, seabream, and turbot) have the ability to digest starch. This activity was present throughout the gut (including the pyloric ceca), so it was possible to establish the ability to hydrolyze carbohydrates (regardless of their feeding habits) in these species [101]. The results obtained by Alarcón et al. [99] point to the existence of a well-developed carbohydrase activity in the gut of seabream, which is confirmed by the presence of carbohydrate activity in the gut (detected in different gut sections, from the stomach to the rectum, but mainly present in the pyloric caeca and the intestine). In our study, the high α-amylase activity in the AA0 diet could be due to the ability of seabream to hydrolyze carbohydrates.

Concerning ALP, a dominant enzyme of the intestinal BB, the activity in fish fed the diets supplemented with HPM (P2 and P1) was significantly higher than in fish fed the control diet (AA0). The activity of the digestive enzymes constitutes a considerable factor in the digestion and absorption of the food, especially those found in the BB section of the intestine which are responsible for the final stages of degradation and assimilation of the food [102] and their activity is considerably regulated by the intraluminal presence of dietary substrates [103,104,105]. ALP is found fundamentally in cell membranes where active transport takes place, which is why it is thought of as a general marker of nutrient absorption [106] and as a marker of intestinal integrity [107]. The functional objective of this enzyme is very far from being completely understood. Nevertheless, it hydrolyzes phosphoester bonds in various organic compounds, such as proteins, lipids, and carbohydrates [108]. The high ALP activity in our study could be due to the content of polypeptides, oligopeptides, small peptides, and free AA present in HPM, and this source of animal protein could have increased the activity of this enzyme.

## 5. Conclusions

The results of this study demonstrate that both dietary supplementation with HPM as well as NT concentrate supplementation in feed for gilthead seabream can improve growth performance. When HPM and NT concentrates are included in low doses (1% and 250 ppm, respectively), an improvement in the nutritive efficiency indices as well as in the digestive enzymatic activities of proteases is observed, especially with P1.

With these results, we hope to generate a contribution to the active search for a wide variety of additives that manage to materialize improvement in the health status and production performance of aquatic animals, for better economic growth of the aquaculture industry given the current increase in costs and low quantity of fish feed.

## Figures and Tables

**Figure 1 animals-13-00205-f001:**
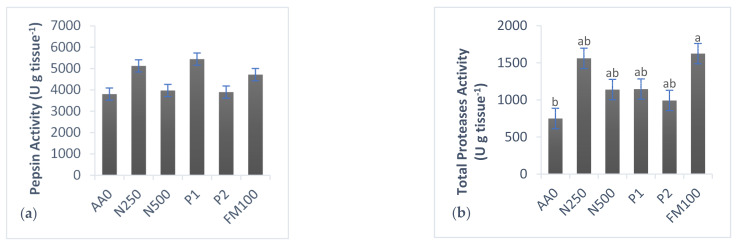
Digestive enzyme activity expressed in U g tissue^−1^ and determined in the gastrointestinal tissue of fish fed the different experimental diets. (**a**) Pepsin activity in stomach tissue; (**b**) total alkaline proteases activity in gut tissue; (**c**) trypsin activity in gut tissue; (**d**) alkaline phosphatase (ALP) activity in gut tissue; (**e**) α-amylase activity in gut tissue. Means of three fish per treatment ± SEM (*n* = 9). Different superscripts indicated differences (*p* < 0.05).

**Table 1 animals-13-00205-t001:** Formulation of experimental diets.

	AA0	N250	N500	P1	P2	FM100
**Ingredients (g kg^−1^)**						
Fishmeal						589
Wheat meal						260
Wheat gluten	290	289.75	289.5	280	270	
Bean meal	44	44	44	44	44	
Soybean meal	182	182	182	182	182	
Pea meal	44	44	44	44	44	
Sunflower meal	181	181	181	181	181	
Fish oil	78	78	78	78	78	38
Soybean oil	78	78	78	78	78	93
Soy lecithin	10	10	10	10	10	10
Vitamin–mineral mix *	10	10	10	10	10	10
Calcium phosphate	38	38	38	38	38	
Taurine	20	20	20	20	20	
Methionine	7	7	7	7	7	
Lysine	10	10	10	10	10	
Arginine	5	5	5	5	5	
Threonine	3	3	3	3	3	
Nucleotide concentrates		0.25	0.5			
Hydrolyzed porcine mucosa				10	20	

* Vitamin and mineral mix (values are g kg^−1^ except those in parenthesis): Premix: 25; Choline, 10; DL-α-tocopherol, 5; ascorbic acid, 5; (PO_4_)_2_Ca_3_, 5. Premix composition: retinol acetate, 1 000 000 IU kg^−1^; calcipherol, 500 IU kg^−1^; DL-α-tocopherol, 10; menadione sodium bisulphite, 0.8; thiamine hydrochloride, 2.3; riboflavin, 2.3; pyridoxine hydrochloride, 15; cyanocobalamine, 25; nicotinamide, 15; pantothenic acid, 6; folic acid, 0.65; biotin, 0.07; ascorbic acid, 75; inositol, 15; betaine, 100; polypeptides 12. Zn, 5; Se, 0.02; I, 0,5; Fe, 0.2; CuO, 15; Mg, 5.75; Co, 0.02; Met, 1.2; Cys, 0.8; Lys, 1.3; Arg, 0.6; Phe, 0.4; Trcp, 0.7; excpt. 1000 g; Dibaq-Diproteg.

**Table 2 animals-13-00205-t002:** Proximate composition of experimental diets.

	Experimental Diets ^1^	FM100
**Proximate composition (% on dry matter)**		
Dry matter (DM) ^2^	94	88
Crude protein (CP)	45.3	45
Total carbohydrates (CHO) ^3^	29.1	23.2
Crude lipid (CL)	20.1	20
Crude fiber (CF)	2.5	1.5
Ash (A)	6.8	10.3
Crude energy (CE) (MJ Kg^−1^) ^4^	17,180	18,827
CP/CE (g MJ^−1^)	38.0	41.9
**Essential amino acids (% on dry matter)**		
Arginine (Arg)	2.68	3.55
Histidine (His)	0.94	1.57
Isoleucine (Ile)	1.67	2.09
Leucine (Leu)	3.06	4.07
Lysine (Lys)	2.40	3.64
Methionine (Met)	1.33	1.41
Threonine (Thr)	1.58	2.17
Valine (Val)	1.77	2.41

^1^ Experimental diets (AA0, N250, N500, P1, and P2) were considered chemically identical. ^2^ Dry matter was expressed as % of wet matter. ^3^ Total carbohydrates were calculated as: CHO = 100 − CP − CL − CF − A. ^4^ Calculated using: 23.9 kJ g^−1^ protein, 39.8 kJ g^−1^ lipid and 17.6 kJ g^−1^ carbohydrate.

**Table 3 animals-13-00205-t003:** Growth and nutritive efficiency indices of gilthead seabream fed the different experimental diets.

	AA0	N250	N500	P1	P2	FM100	SEM
Initial weight (g)	13 ^a^	11 ^ab^	11 ^ab^	10 ^b^	10 ^b^	9.5 ^b^	0.8
Final weight (g)	87 ^e^	111 ^c^	107 ^c^	121 ^b^	99 ^d^	152 ^a^	1.4
SGR (% day^−1^) ^1^	1.59 ^e^	1.74 ^c^	1.71 ^c^	1.81 ^b^	1.65 ^d^	2.09 ^a^	0.01
FI (g 100 g fish^−1^ day^−1^) ^2^	2.14	1.96	2.01	1.87	2.05	2.04	0.10
FCR ^3^	1.93 ^b^	1.60 ^a^	1.66 ^ab^	1.50 ^a^	1.70 ^ab^	1.57 ^a^	0.11
PER ^4^	1.24	1.38	1.32	1.41	1.28	1.41	0.08
S (%) ^5^	89.1	77.3	85.1	84.7	79.6	85.5	4.95

^1^ Specific growth rate (SGR, % day^−1^) = 100 × ln (final weight/initial weight)/days. ^2^ Feed intake (FI, g 100 g fish^−1^ day^−1^) = 100 × feed consumption (g)/average biomass (g) × days. ^3^ Feed conversion ratio (FCR) = feed offered (g)/weight gain (g). ^4^ Protein efficiency ratio (PER) = weight gain (g)/protein offered (g). ^5^ Survival (S, %) = 100 × (final number of fish/initial number of fish). Data are presented as mean ± SEM (*n* = 3). Different superscript letters indicate significant differences amongst treatments (*p* < 0.05).

**Table 4 animals-13-00205-t004:** Biometric indices of seabream fed with different experimental diets.

	AA0	N250	N500	P1	P2	FM100	SEM
CF (g cm^−3^) ^1^	1.60	1.65	1.61	1.67	1.55	1.66	0.04
VSI (%) ^2^	12.01	11.93	12.40	12.68	11.43	12.52	0.97
VFI (%) ^3^	0.61	0.75	0.64	0.59	0.64	0.70	0.16
HSI (%) ^4^	1.79 ^ab^	1.69 ^ab^	1.60 ^b^	1.69 ^ab^	1.68 ^ab^	1.99 ^a^	1.68

^1^ Condition factor (CF, g cm^−3^) = 100 × total weight (g)/total length^3^ (cm). ^2^ Viscerosomatic index (VSI, %) = 100 × visceral weight (g)/fish weight (g). ^3^ Visceral fat index (VFI, %) = 100 × visceral fat weight (g)/fish weight (g). ^4^ Hepatosomatic index (HSI, %) = 100 × liver weight (g)/fish weight (g). Data are presented as least-squares means ± SEM (*n* = 9). Different superscript letters indicate significant differences amongst treatments (*p* < 0.05).

**Table 5 animals-13-00205-t005:** Apparent digestibility coefficients (ADC, %) of crude protein (CP), dry matter (DM), and amino acids (AA) in the gilthead seabream fed different experimental diets.

	AA0	N250	N500	P1	P2	FM100
ADC_CP_	91	93	93	91	93	88
ADC_DM_	63	61	62	73	70	81
ADC_EAA_ ^1^						
Arg	89	90	88	93	90	96
His	93	95	95	95	93	96
Ile	92	94	94	95	92	96
Leu	93	96	95	95	93	96
Lys	94	95	96	96	94	98
Met	96	98	98	98	97	97
Phe	95	96	96	97	95	97
Thr	91	93	93	94	90	96
Val	91	94	94	94	91	96
ADC_NEAA_ ^2^						
Ala	91	92	93	94	90	96
Asp	87	91	92	91	86	92
Aba	89	92	92	93	89	92
Glu	96	97	97	97	96	97
Gly	88	90	90	91	86	93
Pro	95	96	96	96	95	96
Ser	93	95	94	95	93	96
Tyr	95	97	97	97	96	98
AADCEA ^3^	93	95	94	95	93	97

^1^ EAA, essential amino acids; ^2^ NEAA, non-essential amino acids; ^3^ AADCEA, average apparent digestibility coefficient of essential amino acids.

**Table 6 animals-13-00205-t006:** Effect of the different experimental diets on proximal intestine parameters.

	AA0	N250	P1	FM100	SEM
SL (µm)	34.1	36.5	37.2	34.5	1.4
ML (µm)	35.1	35.1	37.5	36.6	1.3
SML (µm)	27.1	23.8	25.3	24.8	1.6
VL (µm)	513.9	470.0	514.7	482.7	34.3
VT (µm)	101.2 ^a^	102.9 ^a^	110.8 ^ab^	114.7 ^b^	4.1
LP (µm)	20.4	21.1	21.2	20.8	1.0
GC	8.5 ^b^	7.6 ^ab^	7.8 ^ab^	6.9 ^a^	1.1

Serous layer (µm), SL; Muscular layer (µm), ML; Submucous layer (µm), SML; Villi length (µm), VL; Villi thickness (µm), VT; Lamina propria (µm), LP; Goblet cells, GC. Data are mean ± SEM (*n* = 9). Different letters indicate that significant differences were observed (*p* < 0.05).

## Data Availability

The data presented in this study are available on request from the corresponding author.

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
