# Peer review of "Effect of Additives Inclusion in Gilthead Seabream (Sparus aurata L.) Diets on Growth, Enzyme Activity, Digestibility and Gut Histology Fed with Vegetable Meals"

_animals, 2023, doi:10.3390/ani13020205_

Round 1
Reviewer 1 Report
Dear authors,
The manuscript has a clear structure and it could be useful to the current literature on the topic of fishmeal replacement in aquaculture nutrition. It also provides the results of using new feed-additive sources in aquaculture studies.
Please find some questions and comments below:
- How long is the digestibility experiment? this information should be added to the manuscript.
- How did the authors add Cr2O3 to the experiment diets? This also needs to add to the text.
- The data presented as mean only. I could not see SEM, please add it in the table ( ex: table 3).
- Explain why the ADC DM is quite low but the ADM protein and AA are pretty high.
Reviewer
Author Response
- Comment reviewer: How long is the digestibility experiment? this information should be added to the manuscript.
- Response: Revised and corrected. Now in the text: “Apparent digestibility experiment was carried out in the same tanks at the end of the growth experiment. The same six diets were used but chromium oxide (50 g kg-1) was added as inert marker. The fish that were not used for the analyses, were left in the tanks and continued to feed with these diets. Faeces collection was done three times a week (to minimize fish suffering) for a period of 21 days. Fish were fed once a day in the morning (05:00) and fecal collection took place 8 h later (13:00)”.
- Comment reviewer: How did the authors add Cr2O3 to the experiment diets? This also needs to add to the text.
- Response: Revised and corrected. Now in the text: “In the vegetable diet (AA0), the protein fraction was supplied entirely by commercial plant ingredients, complying with the minimum requirements of essential amino acids (EAA) established by Peres and Oliva-Teles [49]. Additives were assayed in different dose: nucleotide concentrate in doses of 250 ppm (N250) and 500 ppm (N500), and HPM at levels of inclusion of 1% (P1) and 2% (P2), replacing calcium phosphate. All these diets were isonitrogenous and isoenergetic and had the same chemical composition. In the control diet (FM100), all the protein was provided by FM. 50 g kg-1 of chromium oxide (Cr2O3) was used as an inert marker for the digestibility trial in these same diets. All dry ingredients were uniformly mixed together before adding the liquid ingredients (vegetable and fish oils). Ingredient content is shown in Table 1”.
- Comment reviewer: The data presented as mean only. I could not see SEM, please add it in the table (ex: table 3).
- Response: In the initial document, the SEM values appear in each of the tables. We have verified that this value appears in the current document.
- Comment reviewer: Explain why the ADC DM is quite low but the ADC protein and AA are pretty high.
- Response: It is because the digestibility of carbohydrates and fiber is poor in these experimental diets, while the digestibility of protein is very good. In fact, it is seen that the control diet that has less amount of carbohydrates and fiber has a higher dry matter digestibility.

Reviewer 2 Report
Overall recommendations/comments:
This is an interesting paper and gives information about the benefits of using feed additives on plant-based diets in fish, specifically in gilthead seabream. Fishmeal replacement by other alternative sources is a relevant matter in aquaculture because the price of animal protein is continuously increasing. This study shows promising results and can be applicated to the aquaculture industry. The work supplements several other investigations covering the same topic and many of these papers appear in the references. On the whole, the paper is well-written and considerable thought has been put into interpreting the reported findings. The structure and format are good and the title and abstract give the necessary information about the work
I think the work covered in the manuscript is appropriate for publication in Animals journal after some minor changes.
Introduction:
The introduction is really good, emphasizing the principal subjects of the paper to present the hypothesis.
Material and methods:
The "Material and Methods" section is very good and allows other authors to replicate the experiment. However, I would recommend the authors use another post hoc test to compare means (Tukey, for example), because if SNK is used to compare more than four groups, the familywise error rate increases to unacceptable levels. The Newman-Keuls test has more power than the Tukey test, finding statistical differences that Tukey will not find, but this power comes at a price. When you have more than three groups, SNK will increase the Type I error.
Results:
Results are presented clearly using tables and one figure accompanied by a very descriptive text on the manuscript. Some minor comments:
I missed the p-value on all tables and graphs, it could be useful to show the exact p-value on each parameter.
In table 4, the abbreviation of the hepatosomatic index (HSI) is misspelled.
Lines 332-333: Observing Figure 1, pepsin and alpha-amylase were not significantly affected by diet.
Discussion and conclusion:
The discussion is very good, and the authors show that they know previous research about the subject they are studying. I missed some information about economical points. The results are very good on the parameters studied, but to applicate this to the aquaculture industry, it would be necessary to know how those additives would affect diet prices.
Author Response
- Comment reviewer: The "Material and Methods" section is very good and allows other authors to replicate the experiment. However, I would recommend the authors use another post hoc test to compare means (Tukey, for example), because if SNK is used to compare more than four groups, the familywise error rate increases to unacceptable levels. The Newman-Keuls test has more power than the Tukey test, finding statistical differences that Tukey will not find, but this power comes at a price. When you have more than three groups, SNK will increase the Type I error.
- Response: In all the work we do in our research group we use the Student Newman-Keuls test for the comparision of the means, but if you want us to use the Tukey test we would. It is worth noting that we found no differences between one test and another. For example:
|
AA0 |
N250 |
N500 |
P1 |
P2 |
FM100 |
SEM |
SGR (Newman-Keuls) |
1.54c |
1.74b |
1.72b |
1.84a |
1.66b |
2.09a |
0.02 |
SGR (Tukey HSD) |
1.54c |
1.74ab |
1.72b |
1.84a |
1.66b |
2.09a |
0.02 |
- Comment reviewer: I missed the p-value on all tables and graphs, it could be useful to show the exact p-value on each parameter.
- Response: The exact value of the p-value was not included in the tables since if it is less than or greater than 0.05 there is a statistically significant difference or not, respectively. Also, it was not included so as not to overload the tables with information and the explanatory legend appears at the bottom of each table and figure.
- Comment reviewer: In table 4, the abbreviation of the hepatosomatic index (HSI) is misspelled.
- Response: Revised and corrected.
- Comment reviewer: Lines 332-333: Observing Figure 1, pepsin and alpha-amylase were not significantly affected by diet.
- Response: The sentence has been rewritten: “Activities of digestive enzymes pepsin and α-amylase were unaffected by diets composition, unlike total alkaline proteases, trypsin and alkaline phosphatase which were. Fish fed control diet AA0 had the lowest values enzymatic activity, except for α-amylase (Figure 1).
- Comment reviewer: The discussion is very good, and the authors show that they know previous research about the subject they are studying. I missed some information about economical points. The results are very good on the parameters studied, but to applicate this to the aquaculture industry, it would be necessary to know how those additives would affect diet prices.
- Response: Now in the conclusion: “The results of this study demonstrate that both dietary supplementation with HPM as well as NT concentrates supplementation in feed for gilthead seabream can improve growth performance. When HPM and NT concentrates are included in low doses (1% and 250 ppm, respectively), an improvement in the nutritive efficiency indices as well as in the digestive enzymatic activities of proteases is appreciated, especially with P1. With these results, we hope to generate a contribution to the active search for a wide variety of additives that manage to materialize improvement in health status and production performance of aquatic animals, for better economic growth of the aquaculture industry given the current increase in costs and low quantity of fish feed”.
